# Training and testing of a gradient boosted machine learning model to predict adverse outcome in patients presenting to emergency departments with suspected covid-19 infection in a middle-income setting

**Gordon Ward Fuller**[1]*, **Madina Hasan**[1], **Peter Hodkinson**[2], **David McAlpine**[2], **Steve Goodacre**[1], **Peter A. Bath**[1,3], **Laura Sbaffi**[3], **Yasein Omer**[2], **Lee Wallis**[2], **Carl Marincowitz**[1]

**1** Centre for Urgent and Emergency Care Research (CURE), Health Services Research School of Health and Related Research, University of Sheffield, Sheffield, United Kingdom, **2** Division of Emergency Medicine, University of Cape Town, Cape Town, South Africa, **3** Information School, University of Sheffield, Sheffield, United Kingdom

* g.fuller@sheffield.ac.uk

**Data Availability Statement:** The data used for this study are subject to a data sharing agreement with

## Abstract

COVID-19 infection rates remain high in South Africa. Clinical prediction models may be helpful for rapid triage, and supporting clinical decision making, for patients with suspected COVID-19 infection. The Western Cape, South Africa, has integrated electronic health care data facilitating large-scale linked routine datasets. The aim of this study was to develop a machine learning model to predict adverse outcome in patients presenting with suspected COVID-19 suitable for use in a middle-income setting. A retrospective cohort study was conducted using linked, routine data, from patients presenting with suspected COVID-19 infection to public-sector emergency departments (EDs) in the Western Cape, South Africa between 27th August 2020 and 31st October 2021. The primary outcome was death or critical care admission at 30 days. An XGBoost machine learning model was trained and internally tested using split-sample validation. External validation was performed in 3 test cohorts: Western Cape patients presenting during the Omicron COVID-19 wave, a UK cohort during the ancestral COVID-19 wave, and a Sudanese cohort during ancestral and Eta waves. A total of 282,051 cases were included in a complete case training dataset. The prevalence of 30-day adverse outcome was 4.0%. The most important features for predicting adverse outcome were the requirement for supplemental oxygen, peripheral oxygen saturations, level of consciousness and age. Internal validation using split-sample test data revealed excellent discrimination (C-statistic 0.91, 95% CI 0.90 to 0.91) and calibration (CITL of 1.05). The model achieved C-statistics of 0.84 (95% CI 0.84 to 0.85), 0.72 (95% CI 0.71 to 0.73), and 0.62, (95% CI 0.59 to 0.65) in the Omicron, UK, and Sudanese test cohorts. Results were materially unchanged in sensitivity analyses examining missing data. An XGBoost machine learning model achieved good discrimination and calibration in

the Western Cape Government Department of Health and Wellness, which prohibits further sharing of patient-level or aggregate data. Access to these and related data should be requested directly from this organization (Health. Research@westerncape.gov.za) and is subject to the necessary ethical and organisational approval processes. Full reproducible analysis code is publicly available in the GitHub public repository (https://github.com/gordon-fuller/ICODA). The final XGBoost model is provided in binary format in the supplementary materials.

**Funding:** This work is part of the Grand Challenges ICODA pilot initiative, delivered by Health Data Research UK and funded by the Bill & Melinda Gates Foundation and the Minderoo Foundation. The Provincial Health Data Centre (PHDC), Health Intelligence Directorate, Western Cape Government Health and Wellness acknowledges funding from the United States National Institutes of Health (R01HD080465, U01AI069911), Bill and Melinda Gates Foundation (1164272; 1191327; INV-004657, INV-017293), the Wellcome Trust (203135/Z/16/Z), the United States Agency for International Development (72067418CA00023). PB receives funding from the Health Data Research UK Northern Care Homes Partnership. The funders had no role in study design, data collection and analysis, decision to publish, or preparation of the manuscript.

**Competing interests:** The authors have declared that no competing interests exist.

prediction of adverse outcome in patients presenting with suspected COVID19 to Western Cape EDs. Performance was reduced in temporal and geographical external validation.

## Author summary

The coronavirus disease 2019 (COVID-19) pandemic continues, with ongoing high infection rates. Clinical prediction models are tools that compute the risk of a given patient outcome based on a set of individual characteristics. Such models may be helpful for rapid triage, and supporting clinical decision making, for patients with suspected COVID-19 infection. Machine learning is where a data is provided to a computer algorithm to produce a mathematical model for prediction of future outcomes, such as a clinical prediction model. We developed a machine learning algorithm in many patients with suspected COVID-19 infection from the Western Cape, South Africa during their initial pandemic wave. We then tested it in three other groups of patients: Western Cape patients presenting during the Omicron COVID-19 wave, a UK cohort during the ancestral COVID-19 wave, and a Sudanese cohort during ancestral and Eta waves. We found that the most important features for predicting adverse outcome were the requirement for supplemental oxygen, peripheral oxygen saturations, level of consciousness and age. Our model performed well in Western Cape patients during the initial COVID19 pandemic wave. The model could strongly identify patients who subsequently died or required intensive care treatment. However, performance was reduced in the other settings.

## Introduction

The severe acute respiratory syndrome coronavirus 2 (SARS-CoV-2) emerged in December 2019 and subsequently spread globally, causing the coronavirus disease 2019 (COVID-19) pandemic.[1] To date, South Africa, has experienced four distinct pandemic waves caused by the ancestral Wuhan SARS-CoV-2 strain and subsequent evolutionary variants (Alpha and Beta, Delta, and Omicron). [2,3] Although, much reduced from earlier in the pandemic, infection rates remain high, with approximately 3,000 confirmed cases recorded per week across South Africa in Autumn 2022.[4]

The morbidity and mortality of COVID-19 infection has been attenuated by vaccination, development of natural immunity, and the evolution of less pathogenic variants.[5] However, emergency health care systems in middle-income settings, such as South Africa, remain vulnerable to being overwhelmed due to low vaccine coverage (35% fully vaccinated in October 2022) and emergence of future severe COVID-19 variants.[6] Moreover, healthcare within South Africa emergency systems may be delivered by less experienced clinicians, with restricted access to laboratory or radiological investigations.[7] Clinical prediction models could help risk-stratification of patients presenting to emergency departments (ED) with suspected COVID-19 and support clinical decision making around triage and management decisions for individual patients. Existing models, such as the COVID-specific Pandemic Respiratory Infection Emergency System Triage (PRIEST) score, were developed in high-income settings and may not be generalisable or applicable to less well-resourced settings.[8]

Machine learning is where a data is provided to a computer algorithm to produce a mathematical model for prediction of future outcomes.[9] Machine learning algorithms may have advantages over traditional statistical prediction models, such as logistic regression, in 'big

data' settings, in high signal to noise scenarios, where high-order interactions exist between model inputs, and if continuous model inputs are non-linear.[9] The Western Cape of South Africa has recently developed integration of electronic health care data across prehospital, ED, laboratory, and public health systems. Linkage of these routine data sources provides a unique opportunity to produce a very large study sample amenable to machine learning approaches.

The aim of this study was therefore to develop a model to predict adverse outcome in patients presenting with suspected COVID-19 suitable for use in a middle-income setting. Specific objectives were to train a gradient boosted machine learning model using data from Western Cape EDs, explore which clinical features were most predictive of adverse outcome, and test the model's discrimination and calibration in external validation.

## Methods

### Study design

A retrospective cohort study was conducted to train and test a machine learning model, using previously collected routine electronic data, to predict adverse outcomes in patients with suspected COVID-19, in middle income countries. The study was conducted and reported in accordance with relevant expert guidelines: Transparent reporting of a multivariable prediction model for individual prognosis or diagnosis (TRIPOD), [10] Reporting of studies Conducted using Observational Routinely collected Data (RECORD), [11] and DOME: recommendations for supervised machine learning validation in biology.[12]

### Setting and study populations

The source population for model training (derivation) was patients aged over 16 years presenting with suspected COVID-19 infection to public-sector EDs in the Western Cape, South Africa during the Alpha, Beta and Delta waves of the COVID-19 pandemic (27th August 2020 to 31st October 2021).[3] The subsequent study population comprised consecutive patients presenting to seven hospital EDs contributing to the Hospital Emergency Centre Triage and Information System (HECTIS) data repository. Participating hospitals were from the urban Cape Town metropole district and a single large peri-rural hospital. Patients were included where an ED clinical impression of suspected, or confirmed, COVID-19 infection had been recorded.

Three additional populations were studied for model testing (external validation). A temporal validation sample consisted of Western Cape patients who presented to participating hospitals after the emergence of the Omicron COVID-19 variant (1st November 2021 to 11th March 2022).(3) A geographical, validation sample was derived from the PRIEST mixed prospective and retrospective cohort study that collected data from 70 EDs across 53 sites in the UK during the initial COVID-19 ancestral wave between 26th March and 28th May 2020.[8] A second geographical validation was performed in a retrospective cohort of Sudanese patients presenting to two government referral hospitals in Sudan's most populous region, Khartoum State, between January 2020 and 14th December 2021.[13] This period corresponded to two distinct COVID-19 waves: an initial ancestral wave, and a later Eta variant wave. A base case complete case analysis was performed excluding all cases with any missing values from the training or test data sets.

### Data collection and preparation

The Western Cape data sets were produced by linking information from routinely collected public data sources: ED HECTIS system, National Health Laboratory Services, death

certification and other Western Cape Public Health data sources. Deterministic matching, based on unique patient hospital numbers, was performed by the Western Cape Provincial Health Data Centre (PHDC)). The final data set comprised patient demographics, ED clinical details, COVID-19 status, hospital and critical care admissions, and death during the index COVID encounter. For patients with multiple ED attendances, data were extracted for the first ED attendance and outcomes were assessed up to 30 days from index attendance. Data collection for the UK PRIEST and Sudanese cohort studies have been described in detail in previous publications.[8, 13] Anonymised versions of the PRIEST and Sudan study data were used to derive the geographical external validation cohorts.

Where no comorbidities were recorded, they were assumed not to be present. Implausible physiological variables were set as missing, including systolic blood pressure <50 mmHg, temperature >42 or <25 degrees Celsius, heart rate < 10/minute, peripheral oxygen saturation < 10% and respiratory rate = 0/minute. List-wise deletion of cases with missing feature data was performed for a complete case base case analysis.

## Features and feature engineering

Candidate features were selected *a priori* on the basis of a previous systematic review of COVID-19 outcome predictors suitable for use in lower and middle income countries, [14] previous research, expert opinion within the research team, and availability at ED triage in the Western Cape.[8, 15, 16] The final features considered were: age, sex, presenting symptoms (cough or fever), co-morbidities (heart disease, diabetes, immunosuppression (including HIV), asthma, chronic obstructive pulmonary disease, other chronic respiratory disease, hypertension or pregnancy), first ED recorded physiological parameters (respiratory rate, pulse rate, systolic blood pressure, level of consciousness, peripheral oxygen saturations) and requirement for supplemental oxygen in the ED. Comorbidities were one-hot encoded, with asthma, chronic obstructive pulmonary disease, and other chronic respiratory disease grouped into a single feature. Level of consciousness, recorded using the ACVPU scale in Western Cape data, was pre-processed to a numeric AVPU scale (confusion, grouped with verbal) to ensure consistency across data sets. Continuous physiological features were not transformed, and no other feature engineering was performed. No feature selection was performed prior to machine learning. All features were available for the Western Cape and PRIEST data. A restricted range of features was available in Sudanese data, with no information available for temperature, immunosuppression, or level of consciousness.

## Label

The label was a composite, binary, adverse outcome of either intubation or non-invasive ventilation in the ED on index attendance, Intensive Care Unit (ICU) admission or inpatient death up to 30 days from index attendance. This was comparable to the PRIEST study primary outcome (used in the geographical external validation sample) of death or organ support (respiratory, cardiovascular, or renal) at 30 days.(8) Outcome in the Sudanese data was intubation or non-invasive ventilation in the ED, High/Intensive Care Unit (HDU/ICU) admission or inpatient death. The model aimed to provide both good discrimination and well calibrated predictions of adverse outcome, rather than label classification per se, therefore methods to address class imbalance were not performed.[17]

## Model training

Supervised machine learning was performed using an ensemble, decision tree-based, gradient boosting algorithm, implemented using the XGBoost framework.[18] This approach was

implemented over alternative algorithms due to its scalability to large datasets, flexibility in capturing non-linear relationships and interactions, and favourable predictive performance compared to other algorithms.[19] Training was initially performed using default parameters, with regularization and early stopping rules defined to reduce variance and mitigate over-fitting. Key hyperparameters (number of decision trees, learning rate, and tree depth) were tuned using 5-fold cross validation of the training data, across a manually selected range of parameter values, aiming to optimise model discrimination. To facilitate out of sample prediction, conservative hyperparameter values were favoured in the absence of significant gains in model performance.

The relative importance of each feature in the base case model was evaluated by calculating the average training loss reduction gained across all occasions it was used for decision tree splitting. A waterfall chart, showing the additive contribution of each individual feature for label prediction, was constructed for illustrative cases to aid model interpretation. Machine learning was carried out with the xgboost library in R 4.1.2 (R Core Team, 2021) using the R Studio interface.[20] The DALExtra R package was used to construct waterfall charts. Models were independently developed using the same methods by a second data analyst in Python using the Scikit-learn package (version 0.24.1) using Python (Python Software Foundation. Python Language Reference, version 3.8.8).

## Model testing

Internal validation, using a random 80:20 train-test ratio split, was performed. The model was then re-trained on the whole training dataset, and apparent validation assessed. External validation was evaluated by application of the final model to the Western Cape Omicron period, UK PRIEST, and Sudanese external validation cohorts with adverse outcome probability calculated for each case. Model discrimination was assessed through receiver-operating characteristic (ROC) curves and calculating the area under the ROC curve (C-statistic).[21] Calibration in the large was evaluated by comparing the average predicted risk to the average observed risk. Calibration plots were constructed to compare predicted to observed risks across deciles of predicted outcome probability. Calibration plot slope and intercept (weak calibration) and Locally Weighted Scatterplot Smoothing (LOWESS, moderate calibration) were also evaluated.[22] Diagnostic parameters (accuracy, precision, recall, negative predictive value, and specificity), and the proportion of cases with adverse outcome, were calculated and presented graphically for different model probability thresholds to inform clinical management decisions.[23] Discrimination was computed using the pROC library in R 4.1.2 (R Core Team, 2021). Calibration metrics and plots were calculated using the pmcalplot package in Stata 17.0 (StataCorp. 2021. College Station, TX: StataCorp LLC).

## Secondary analyses

Case-wise and variable-wise missing data patterns were examined and the influence of missing data was explored using 3 approaches: deterministic imputation (single imputation within normal ranges defined by the South Africa Triage Early Warning Score (TEWS) score), [24] multiple imputation approaches (data assumed to be missing at random, chained equations, 5 imputations, model predictions averaged across datasets), [25] and using the in-built XGBOOST missing data algorithm (based on surrogate decision tree splits).[18] For simplicity and increased usability, an alternative model was also developed using categorised physiological variables, engineering features according to thresholds used by (TEWS) score in complete case data.[24] To provide an indication of likely performance in future waves, irrespective of vaccination prevalence and variant dominance, an additional model was trained on Western

Cape data across the alpha, delta, and omicron waves. A random 80:20 train/test split of the pooled data was used, with modelling and validation otherwise proceeding as described for the base case model.

## Sample size

The training sample size was fixed based on a census sample of patients in the Western Cape recorded on the HECTIS during the study period. There were 282,051 patients in this cohort, with over 100 outcomes per model parameter. Test sample size was also fixed based on the size of Western Cape Omicron wave, PRIEST and Sudanese data sets. However, assuming an outcome prevalence of 10% and c-statistic of 0.75, external validation would require only 6,420 cases to provide measurement of the area under the operating characteristic curve with a standard error of 0.01.[26]

## Ethics

Use of routinely collected electronic health care records from the Western Cape for the derivation of the development and Omicron cohorts for this study was approved by the University of Cape Town Human Research Ethics Committee (HREC 594/2021), and the Western Cape Health Research Committee (WC_202111_034). Analysis of Sudanese data was approved by the University of Cape Town Human Research Ethics Committee (HREC 594/2021), the Western Cape Health Research Committee (WC_202111_034) and Khartoum State ministry of health. As all data were de-identified at source before being provided to the research team the need for patient consent was waived. Data collection for the UK validation cohort was first approved by the Northwest—Haydock Research Ethics Committee on 25 June 2012 (reference 12/NW/0303) and on the updated PRIEST study on 23rd March 2020. The Confidentiality Advisory Group of the Health Research Authority granted approval to collect data without patient consent in line with Section 251 of the National Health Service Act 2006.

## Patient and Public Involvement (PPI)

A community advisory board comprising eight community members affected by COVID (infected themselves or immediate family infected/ hospitalised) was purposively recruited by an experienced community liaison officer to achieve representation across the Western Cape population. Through several meetings, the community advisory board were able to influence study planning and conduct. Implementation of machine learning and algorithmic (un)fairness were considered to ensure acceptability and avoid minority group disparities.

## Results

### Study sample

A total of 305,564 patients aged over 16 years presented to participating Western Cape hospitals with suspected COVID-19 during the Alpha, Beta, and Delta waves between 27th August 2020 and 31st October 2021. Of these, 282,051 (92.3%) cases had complete data available and were included in the base case training dataset. The prevalence of 30-day adverse outcome was 4.0%, and 74,580 patients (24.4%) had a diagnosis of COVID confirmed by PCR testing. There were 140,520 patients in the Omicron wave test (external validation) cohort, of whom 130,407 (92.8%) had complete data. The PRIEST test (external validation) cohort comprised 20,698 patients, of whom 18,960 (91.6%) had complete data. The Sudanese test (external validation) cohort comprised 2,583 patients, of whom 1,290 (49.9%) had complete data. Prevalence of adverse outcome was 1.98%, 22.1%, 35.7% in the Omicron wave, PRIEST and Sudanese

cohorts respectively. Table 1 summarises characteristics of the Western Cape data. Derivation of the Western Cape, PRIEST, and Sudanese cohorts; and comparison of patient characteristics across train and test cohorts is provided in the supplementary materials (S1–S3 Fig; and S1–S3 Text).

## Trained model

The base case XGBOOST model hyper-parameters following tuning are presented in Table 2. The most important features for predicting adverse outcome in the final model were the requirement for supplemental oxygen, peripheral saturations, level of consciousness and age (Fig 1 and 2). Apparent validation revealed excellent discrimination (C-statistic 0.91, 95% CI 0.90 to 0.91). This was unchanged on internal validation using a split train-test sample (C-statistic 0.91, 95% CI 0.90 to 0.91, Fig 3), with excellent calibration also apparent (CITL of 1.05, Fig 3). The final base-case model saved in XGBoost-internal binary format is available in the supplementary materials (S1 Data).

The final model showed good discrimination in the Western Cape Omicron test cohort (C-statistic 0.84, 95% CI 0.83 to 0.85, Fig 3); however calibration was sub-optimal with overprediction of adverse outcome across all risk subgroups (CITL of 1.7, Fig 3). Discrimination was reduced in the UK PRIEST cohort (c-statistic 0.72, 95% CI 0.71 to 0.73, Fig 3) with underprediction of adverse outcome (CITL of 0.68, Fig 3). Model discrimination was lower in the Sudanese cohort (C-statistic 0.62, 95% CI 0.59 to 0.65, Fig 3) with under-prediction of adverse outcome for low and moderate risk patients. Estimated diagnostic accuracy (recall, specificity, NPV, precision) at different model probability thresholds across the train and test populations are presented in Fig 4, with tables available in the supplementary materials (S4–S7 Text).

## Secondary analyses

Case- and variable-wise missing data patterns are presented in the supplementary materials for training and testing data (S4–S10 Figs). Results were not substantively changed in secondary analyses exploring different missing data mechanisms. (. On internal validation, C-statistics ranged from 0.891 to 0.892 across deterministic, surrogate split, and multiple imputation analyses, and calibration plots were not significantly changed from the complete data base case analysis. Discrimination and calibration metrics were similarly unchanged in missing data secondary analyses in the Western Cape Omicron, UK PRIEST, and Sudanese test cohorts (S11–S13 Fig). The alternative model, using categorised physiological features in complete case data, achieved a minimal reduction in discrimination (C-statistic 0.90, 95% CI 0.89–0.91), and similar calibration (CITL 1.05) compared to the base case continuous model (S14 Fig). The model trained on Western Cape data across all variants and time periods, demonstrated similar discrimination (C-statistic 0.91, 95% CI 0.90–0.91) and calibration (CITL 1.07) metrics compared to the base case model (S15 Fig).

## Discussion

### Summary of results

An XGBoost machine learning model was trained in patients with suspected COVID19 presenting to Western Cape public hospitals during the Alpha/Beta/Delta pandemic waves. The most important features for predicting adverse outcome were the requirement for supplemental oxygen, peripheral oxygen saturations, level of consciousness and age. Internal validation using a split-test sample revealed excellent discrimination (C-statistic 0.91, 95% CI 0.90 to 0.91) and calibration (CITL of 1.05). The model achieved C-statistics of 0.84 (95% CI 0.84 to

**Table 1. Characteristics of Western Cape Alpha/Beta/Delta wave study participants.**

| Characteristic | Statistic/level | Adverse outcome | No adverse outcome | Total |
|---|---|---|---|---|
| Age (years) | N | 12,610 (4.1%) | 292,954 (95.9%) | 305,564 |
| | Mean (SD) | 56.5 (17.5) | 43.2 (17.1) | 43.7 (17.3) |
| | Median (IQR) | 59 (43, 70) | 40 (29, 56) | 41 (29, 57) |
| | Range | 16 to 105 | 16 to 110 | 16 to 110 |
| Sex | Male | 6,670 (52.9%) | 151,294 (51.6%) | 157,964 (51.7%) |
| | Female | 5,940 (47.1%) | 141,660 (48.4%) | 147,600 (48.3%) |
| Comorbidities | Asthma/COPD | 2,220 (17.6%) | 42,590 (14.5%) | 44,810 (14.7%) |
| | Other Chronic respiratory disease | 69 (0.6%) | 649 (0.2%) | 718 (0.2%) |
| | Diabetes | 5,256 (41.7%) | 51,622 (17.6%) | 56,878 (18.6%) |
| | Hypertension | 5,863 (46.5%) | 80,099 (27.3%) | 85,962 (28.1%) |
| | Immunosuppression (HIV) | 1,553 (12.3%) | 50,824 (17.4%) | 52,377 (17.1%) |
| | Heart Disease | 4,560 (36.2%) | 53,664 (18.3%) | 58,224 (19.1%) |
| | Pregnant | 62 (0.5%) | 1,915 (0.7%) | 1,977 (0.7%) |
| AVPU | Missing | | | 9,229 (3.0%) |
| | Alert | 9,159 (72.6%) | 264,460 (90.3%) | 273,619 (89.6%) |
| | Voice | 288 (2.3%) | 3,682 (1.3%) | 3,970 (1.3%) |
| | Confused | 617 (4.9%) | 11,661 (4%) | 12,278 (4%) |
| | Pain | 593 (4.7%) | 2,202 (0.8%) | 2,795 (0.9%) |
| | Unresponsive | 1,355 (10.8%) | 2,318 (0.8%) | 3,673 (1.2%) |
| Systolic BP | Missing | | | 10,389 (3.4%) |
| (mmHg) | N | 11,801 | 283,374 | 295,175 |
| | Mean (SD) | 130.9 (29.4) | 131.9 (25.5) | 131.8 (25.6) |
| | Median (IQR) | 128 (110,146) | 129 (115,145) | 129 (115,144) |
| | Range | 50 to 289 | 50 to 300 | 50 to 300 |
| Pulse rate | Missing | | | 9,995 (3.3%) |
| (beats/min) | N | 11, 858 | 283,711 | 295,569 |
| | Mean (SD) | 98.8 (23.4) | 93.5 (21) | 93.7 (21.1) |
| | Median (IQR) | 98 (83,113) | 92 (79, 106) | 92 (79,107) |
| | Range | 11 to 300 | 10 to 300 | 10 to 300 |
| Respiratory rate | Missing | | | 9,969 (3.3%) |
| (breaths/min) | N | 11,850 | 283,745 | 295,595 |
| | Mean (SD) | 22.2 (6.7) | 18.6 (4.1) | 18.8 (4.3) |
| | Median (IQR) | 20 (18,25) | 18 (16,20) | 18 (16,20) |
| | Range | 2 to 60 | 1 to 60 | 1 to 60 |
| Oxygen | Missing | | | 27, 781 (6.2%) |
| saturations | N | 11,634 | 274,409 | 286,043 |
| | Mean (SD) | 89.7 (12) | 96.2 (5.5) | 96 (6) |
| | Median (IQR) | 94 (86, 98) | 98 (96, 99) | 97 (95, 99) |
| | Range | 10 to 100 | 10 to 100 | 10 to 100 |
| Supplemental | Missing | | | 18,794 (6.2%) |
| oxygen | 1 (air) | 6,254 (49.6%) | 254,399 (86.8%) | 260,653 (85.3%) |
| administration | 2 (40% O2) | 346 (2.7%) | 5,360 (1.8%) | 5,706 (1.9%) |
| | 3 (28% O2) | 8 (0.1%) | 222 (0.1%) | 230 (0.1%) |
| | 4 (Nasal prongs) | 1,123 (8.9%) | 8,389 (2.9%) | 9,512 (3.1%) |
| | 5 (FM neb) | 27 (0.2%) | 571 (0.2%) | 588 (0.2%) |
| | 6 (rebreather mask) | 1,538 (12.2%) | 5,199 (1.8%) | 6,737 (2.2%) |
| | 7 (nasal prongs and rebreather mask) | 368 (2.9%) | 884 (0.3%) | 1,252 (0.4%) |

(*Continued*)

**Table 1.** (Continued)

| Characteristic | Statistic/level | Adverse outcome | No adverse outcome | Total |
|---|---|---|---|---|
| | 8 intubated | 1,917 (15.2%) | 0 | 1,917 (0.6%) |
| | 9 NIV | 165 (1.3%) | 0 | 165 (0.1%) |
| Temperature (˚C) | Missing | | | 9,252 (3%) |
| | N | 12, 010 | 284,302 | 296,312 |
| | Mean (SD) | 36.4 (1.3) | 36.3 (0.8) | 36.4 (0.9) |
| | Median (IQR) | 36.4 (35.9, 37) | 36.3 (36, 36.7) | 36.3 (36, 36.7) |
| | Range | 25 to 41 | 25 to 42 | 25 to 42 |
| Cough | Missing | | | 41,524 (29.6%) |
| | Present | 557 (4.4%) | 8,538 (2.9%) | 9,095 (3%) |
| Fever | Missing | | | 93,962 (30.8%) |
| | Present | 178 (1.4%) | 2,829 (1%) | 3,007 (1%) |
| COVID PCR | Positive | 10,908 (86.5%) | 63,672 (21.7%) | 74,580 (24.4%) |
| Hospital admission | ICU | 1,527 (12.1%) | 0 | 1,527 (0.5%) |
| Death | Within 30 days contact | 9,711 (77%) | 0 | 9,711 (3.2%) |

0.85), 0.72 (95% CI 0.71 to 0.73) and 0.62, (95% CI 0.59 to 0.65) in the Western Cape Omicron wave, UK, and Sudanese external validation cohorts. Calibration was sub-optimal with over-prediction of adverse outcome across all risk subgroups in Omicron cases (CITL of 1.7); and underprediction of adverse outcome in UK cases with ancestral COVID19 infection and Sudanese patients with ancestral or Eta variant infection (CITL of 0.68 and 0.61 respectively). Results were not substantively changed in extensive sensitivity analyses exploring different missing data mechanisms.

## Interpretation

Clinical prediction models rarely out-perform the clinical judgement of experienced clinicians.[27, 28] Western Cape EDs demonstrated excellent diagnostic performance for management of COVID19 during the study period, admitting only 14.7% of patients as inpatients,

**Table 2. Base case model parameterisation.**

| Parameter | Definition | Value used |
|---|---|---|
| booster | Linear or tree-based model | gbtree |
| nrounds | Number of decision trees | 50 |
| eta | Learning rate | 0.1 |
| gamma | Minimum loss reduction needed to partition in each tree. | 1 |
| alpha | L2 Regularization | 1 |
| max_depth | Maximum depth of decision trees | 6 |
| min_child_weight | Minimum sum of weights of all observations required in a child | 1 |
| sub_sample | Proportion of training data sub-sampled for each additional decision tree | 1 |
| colsample_bytree | Proportion of features sub-sampled for each additional decision tree | 1 |
| scale_pos_weight | Balance of weighting of positive and negative cases | 1 |
| objective | Loss function to be minimised | binary:logistic |
| eval_metric | Metric to be used for hyper-parameter tuning | auc: Area under the curve |
| seed | Random number seed | 12345 |

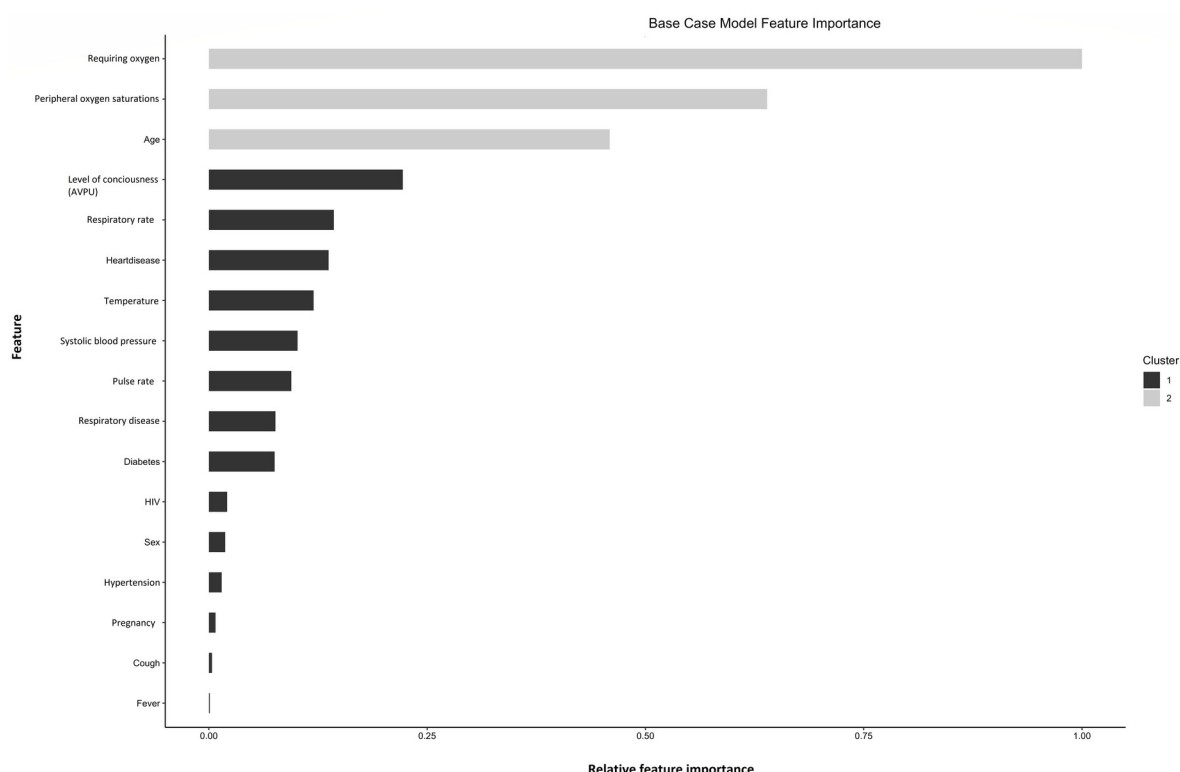

**Fig 1. Relative feature importance plot.**

with a risk of false negative triage of around 1%.[29] Based on internal validation results, a model probability cut point of 8% predicted risk would result in a similar admission rate of 13.1% and a negative predictive value of 98.7%. Despite the favourable accuracy of clinical

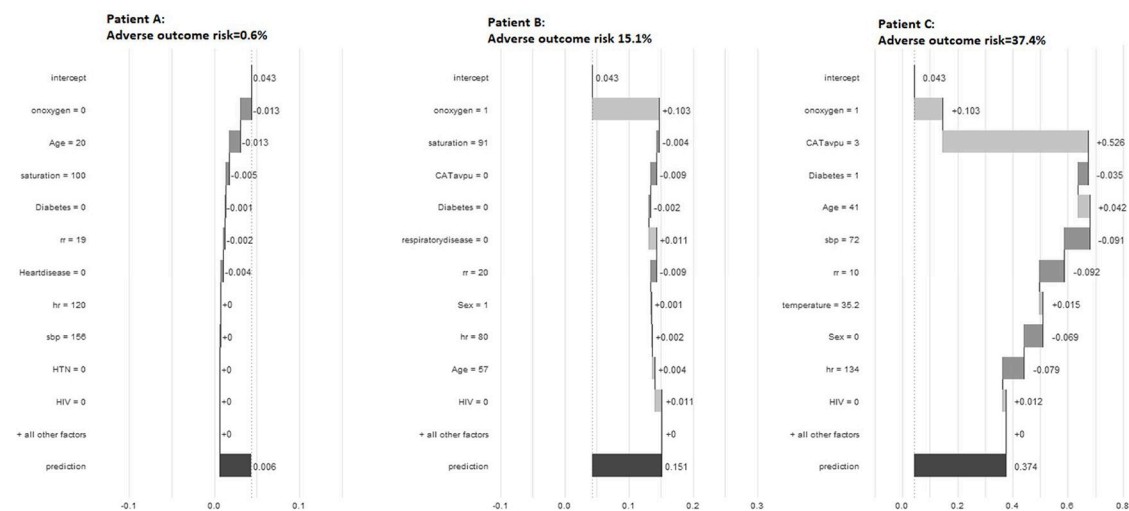

**Fig 2. Waterfall plot, interpretating the base case XGBoost Model for 3 illustrative training cases.** The model intercept of 0.043 represents the average probability of adverse outcome in the Western Cape Alpha/Beta/Delta wave. The additive impact of each feature on probability of adverse outcome for each model feature is shown for 3 representative cases with final predicted probabilities of adverse outcome of 0.006, 0.151, and 0.374. rr: respiratory rate; sbp: systolic blood pressure, hr:heart rate; HTN: hypertension; HIV: immunocompromised; CATavpu: AVPU response, 3 = unresponsive.

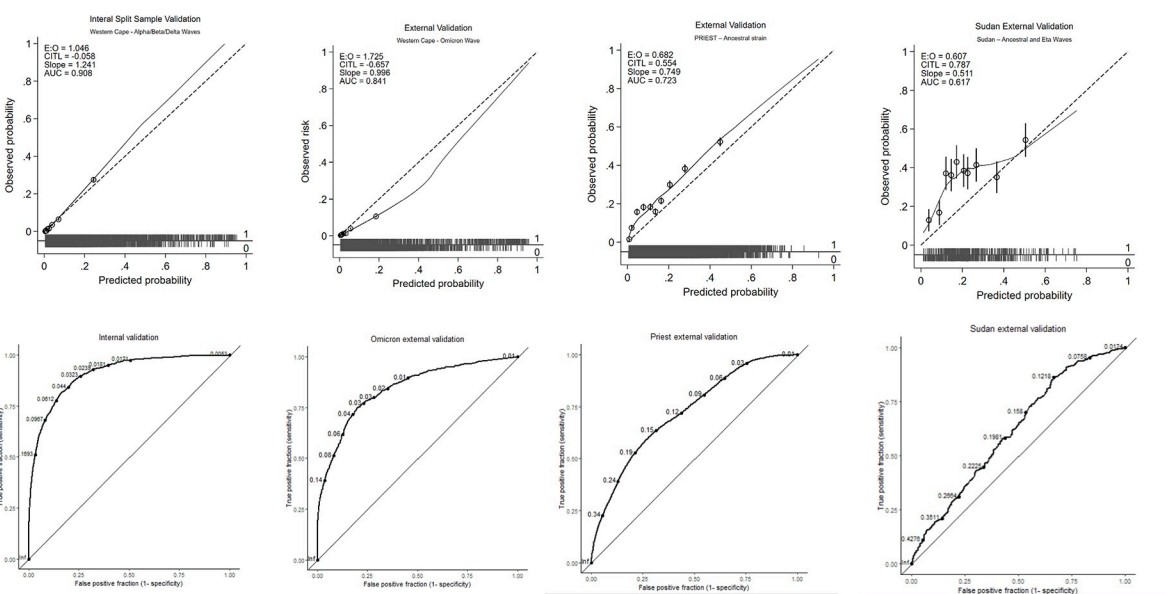

**Fig 3. Calibration plots (top panel) and receiver operating characteristic curves (bottom panel) for test cohorts.** ROC curves are labelled with 10 representative probability thresholds.

gestalt, prediction models are reproducible, objective, may be used by less experienced health care workers, and could save time during periods of increased demand.

The reduced model performance across test cohorts could be explained by inherent study biases, model overfitting, spectrum effects, conflicting associations with individual elements of the composite label (death and organ support), or calibration drift. The wide range of adverse outcome risk (1.98% Omicron wave, to 35.7% ancestral/Eta strain) observed across different pandemic waves clearly demonstrates the potential for spectrum effects, where performance of tests varies across settings with differing disease prevalence. Typically, for any given model

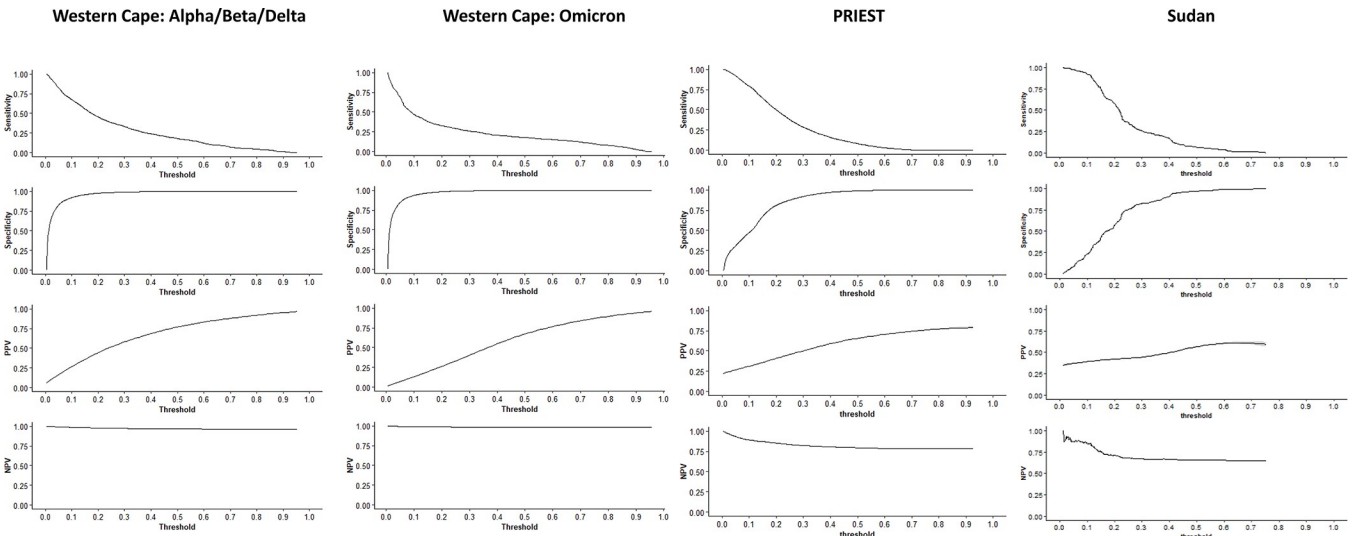

**Fig 4. Threshold plots presenting diagnostic accuracy metrics across varying model probability cut-points for test cohorts.** Sensitivity (recall); NPV: Negative predictive value; PPV: positive predictive value (precision).

threshold, sensitivity will fall, and specificity will increase, at lower prevalence.[30] The high proportion of adverse outcome in the Sudanese test cohort may reflect differing patient demographics, increased SARS-CoV-2 virulence, or inclusion of more severe cases from tertiary referral hospitals. However, the primary reasons for lack of model transportability to non-Western Cape settings are likely to be differences in population characteristics and variation is measurement of features and labels across data sets.

The model label of adverse outcome from death or ICU admission/organ support has face validity for guiding clinical management decision in the ED. However, it is important to note that the model may predict differentially across the individual outcomes comprising the composite endpoint.[31] Differences in the relative proportions of death and organ support across test cohorts could therefore explain differential model performance. The previously published PRIEST score demonstrated better prediction for death than critical care requirement, [8] implying that the current model should be restricted for determining hospital admission only, rather than guiding escalation of care decisions. Additionally, caution may be required when interpreting model outputs in advance age, where the underlying hazard of mortality may dominate prediction.

Calibration drift occurs when deploying models in non-stationary clinical scenarios, [32] where differences arise over time between the training and test populations to which the model is applied. Predicting outcome in patients with suspected COVID19 is a highly dynamic situation with multiple potential changes in data collection, patient case mix, vaccination coverage, and clinical decision-making. Adaptive mutations in the SARS-CoV-2 genome can alter the virus's pathogenic potential, influencing transmissibility, virulence, and vaccination effectiveness.[33] The base case model, trained using Alpha/Beta/Delta data, retained favourable discrimination in patients with Omicron variant, but systematically overestimated individual risk adverse outcome. Updated model training and re-calibration will likely be required as the COVID19 pandemic evolves, and future variants of concern emerge in South Africa.

## Comparison to other literature

There is a paucity of research investigating risk-stratification scores for patients presenting to EDs with suspected COVID19 in middle-income settings. The Nutri-CoV score was developed and internally validated in Mexico using data from the ancestral Wuhan strain of the pandemic.[34] A smaller number of features were included in that model (age, comorbidities, peripheral oxygen saturations, respiratory rate, pneumonia), with lower discrimination for adverse outcome achieved than the current study (C-statistic 0.797). The study sample of RT-PCR confirmed cases may limit generalisability to use in undifferentiated ED patients. Moreover, the inclusion of diagnosis of pneumonia in the score, requiring imaging or advanced clinical skills, may reduce relevance in less well-resourced settings.

Our research group has previously developed similar risk-stratification scores using statistical modelling, rather than machine learning. Multivariable logistic regression with Least Absolute Shrinkage and Selection Operator (LASSO) and fractional polynomials achieved a slightly lower C-statistic of 0.87 (95% CI 0.866 to 0.874) and (CITL) of -0.017 (95%CI -0.043 to 0.009) on internal validation in Western Cape Alpha/Beta/Delta data, compared to the machine learning model reported here.[35] Discrimination on external validation was lower in the Omicron (C-statistic 0.79, 95% CI: 0.79 to 0.80) and Sudanese data (c-statistic 0.53, 95% CI: 0.53 to 0.54), but higher in the UK PRIEST, test data (C-statistic 0.79, 95% CI: 0.79 to 0.80). Machine learning is a trade-off between bias and variance, and despite the large sample size and use of regularization, there is a risk of over-fitting to the Western Cape data, which may explain the differential performance across test cohorts compared to a more conservative statistical modelling strategy.[36]

## Limitations

This study has several strengths including large sample size, adherence to established prediction modelling principles, external validation, assessment of calibration, and exploring model interpretability.[10] However, there are potential limitations. There is a risk of selection bias from incomplete identification of patients with suspected COVID19 infection, inaccurate linking of health records, and incomplete outcome ascertainment.[37] Furthermore, list-wise deletion of missing data, and multiple imputation, could result in systematic error if data is not missing completely at random, or missing at random.[38] The use of routine data, not primarily intended for research purposes, could also result in information bias from measurement error and incomplete ascertainment of deaths.[11]

## Generalisability

The external validity of any COVID19 prediction model will depend on circulating COVID-19 variant, population vaccination status, clinical setting, and underlying patient demographics. The model's inclusion of basic patient characteristics and vital signs should ensure the model is transportable to other middle-income settings. However, the requirement for pulse oximetry may limit application in lower income settings. South Africa is an upper middle-income country, with large wealth disparities, a mixed state-private health economy, and a high prevalence of HIV.[39] Generalisability of model predictions to other middle-income settings with differing characteristics therefore requires caution.

## Clinical and research implications

Despite increasing numbers of published machine learning models, very few are implemented into clinical practice.[40] Furthermore, there is little experience of deploying machine learning models into clinical practice outside high-income settings.[41] Independent external validation, impact studies, and qualitative work to explore acceptability are therefore recommended prior to any introduction of the current model into clinical use. Western Cape emergency medical services use an electronic patient record with the functionality to incorporate electronic decision support, potentially facilitating translation into clinical practice. Model operationalisation as a smartphone is an alternative strategy that could aid usability once regulatory requirements are met. The 'black box' nature of artificial intelligence, with a patient-level prediction provided without any explanation or rationale, is a major barrier to uptake of machine learning models.[9] Using explainable machine learning tools, such as waterfall charts, could help with future model implementation. [42]

## Conclusions

An XGBoost machine learning model was trained and achieved good discrimination and calibration in prediction of adverse outcome in patients presenting to Western Cape EDs with suspected COVID19 infection. Performance was reduced in temporal and geographical external validation. Independent external validation, impact studies, and qualitative work to explore acceptability are recommended prior to any introduction of the current model into clinical use. Updated model training and re-calibration will likely be required as the COVID19 pandemic evolves, and future variants of concern emerge.

## Supporting information

**S1 Fig. Flow diagram showing derivation of complete case study sample for base case model training in Western Cape data and external validation.**
(TIF)

**S2 Fig. Flow diagram showing derivation of complete case study sample for base case model testing in PRIEST data.**
(TIF)

**S3 Fig. Flow diagram showing derivation of complete case study sample for base case model testing in Sudanese data.**
(TIF)

**S4 Fig. Variable-wise missing data in training data (Western Cape Alpha/Beta/Delta wave).**
(TIF)

**S5 Fig. Case-wise missing data patterns in training data (Western Cape Alpha/Beta/Delta wave).** N = 305,564. 7.7% of cases (n = 23,513) had missing data.
(TIF)

**S6 Fig. Variable-wise missing data in test data (Western Cape Omicron wave).**
(TIF)

**S7 Fig. Case-wise missing data patterns in test data (Western Cape Omicron wave).** N = 140,520. 7.2% of cases (n = 10,113) had missing data.
(TIF)

**S8 Fig. Variable-wise missing data in test data (PRIEST).**
(TIF)

**S9 Fig. Case-wise missing data patterns in test data (PRIEST).** N = 20,698. 6.2% of cases (n = 1,291) had missing data.
(TIF)

**S10 Fig. Variable-wise missing data in test data (Sudan).**
(TIF)

**S11 Fig. Missing data secondary analysis—XGBoost inbuilt missing data method.** Calibration plots (bottom panel) and receiver operating characteristic curves (top panel) for test cohorts. ROC curves are labelled with 10 representative probability thresholds.
(TIF)

**S12 Fig. Missing data secondary analysis—Single deterministic imputation.** Calibration plots (bottom panel) and receiver operating characteristic curves (top panel) for test cohorts. ROC curves are labelled with 10 representative probability thresholds.
(TIF)

**S13 Fig. Missing data secondary analysis—Multiple imputation.** Calibration plots (bottom panel) and receiver operating characteristic curves (top panel) for test cohorts. ROC curves are labelled with 10 representative probability thresholds
(TIF)

**S14 Fig. Secondary analysis–Complete case categorised model.** Calibration plots (bottom panel) and receiver operating characteristic curves (top panel) for test cohorts. ROC curves are

labelled with 10 representative probability thresholds.
(TIF)

**S15 Fig. Secondary analysis–Complete case continuous model trained on all Western Cape data (Alpha, Beta, Delta, Omicron waves).** Calibration plots (bottom panel) and receiver operating characteristic curves (top panel) for test cohorts. ROC curves are labelled with 10 representative probability thresholds. NB. Sudanese data not available for external validation of this model, as analysis performed after end of data sharing agreement.
(TIF)

**S1 Data. Final XGBoost model.**
(MODEL)

**S1 Text. Characteristics of Western Cape Omicron wave test cohort.**
(DOCX)

**S2 Text. Population characteristics UK PRIEST test cohort.**
(DOCX)

**S3 Text. Population characteristics Sudan test cohort.**
(DOCX)

**S4 Text. Diagnostic accuracy at different base case model thresholds in Western Cape Alpha/Beta/Delta wave training data.**
(DOCX)

**S5 Text. Diagnostic accuracy at different base case model thresholds in thresholds in Western Cape Omicron wave test data.**
(DOCX)

**S6 Text. Diagnostic accuracy at different base case model thresholds in PRIEST test data.**
(DOCX)

**S7 Text. Diagnostic accuracy at different base case model thresholds in thresholds in Sudanes test data.**
(DOCX)

## Acknowledgments

This work uses data provided by patients as part of their care and support and the authors wish to recognise the Western Cape Government Health and Wellness (WCGHW) for their contribution of the data that made this research possible, specifically Nesbert Zinyakatira and the team from the Provincial Health Data Centre, Health Impact Assessment Directorate, Western Cape Government Health; and Dr Moosa Parak and the HECTIS team. We further acknowledge and thank the National Health Laboratory Service of South Africa, for their contribution to the study through the provision of the digitised laboratory results accessed through the Provincial Health Data Centre. Dr Laura Sutton (University of Sheffield) conducted the statistical analysis for the UK PRIEST study and provided a template for the statistical analyses.

## Author Contributions

**Conceptualization:** Gordon Ward Fuller, Madina Hasan, Peter Hodkinson, Steve Goodacre, Peter A. Bath, Carl Marincowitz.

**Data curation:** Gordon Ward Fuller, Madina Hasan.

**Formal analysis:** Gordon Ward Fuller, Madina Hasan, Carl Marincowitz.

**Funding acquisition:** Gordon Ward Fuller, Madina Hasan.

**Investigation:** Gordon Ward Fuller, Madina Hasan, Carl Marincowitz.

**Methodology:** Gordon Ward Fuller, Madina Hasan, Peter Hodkinson, David McAlpine, Steve Goodacre, Peter A. Bath, Laura Sbaffi, Yasein Omer, Lee Wallis, Carl Marincowitz.

**Project administration:** Gordon Ward Fuller, Madina Hasan, Peter Hodkinson, David McAlpine, Peter A. Bath, Carl Marincowitz.

**Resources:** Gordon Ward Fuller.

**Supervision:** Peter A. Bath, Lee Wallis, Carl Marincowitz.

**Validation:** Gordon Ward Fuller.

**Writing – original draft:** Gordon Ward Fuller, Madina Hasan, Peter Hodkinson, David McAlpine, Steve Goodacre, Peter A. Bath, Laura Sbaffi, Yasein Omer, Lee Wallis, Carl Marincowitz.

**Writing – review & editing:** Gordon Ward Fuller, Madina Hasan, Peter Hodkinson, David McAlpine, Steve Goodacre, Peter A. Bath, Laura Sbaffi, Yasein Omer, Lee Wallis, Carl Marincowitz.

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
