## [Decision Letter · Decision Letter 0]

24 Feb 2023

PDIG-D-23-00004

TRAINING AND TESTING OF A GRADIENT BOOSTED MACHINE LEARNING MODEL TO PREDICT ADVERSE OUTCOME IN PATIENTS PRESENTING TO EMERGENCY DEPARTMENTS WITH SUSPECTED COVID-19 INFECTION IN A MIDDLE-INCOME SETTING

PLOS Digital Health

Dear Dr. Fuller,

Thank you for submitting your manuscript to PLOS Digital Health. After careful consideration, we feel that it has merit but does not fully meet PLOS Digital Health's publication criteria as it currently stands. Therefore, we invite you to submit a revised version of the manuscript that addresses the points raised during the review process.

Please submit your revised manuscript within 60 days Apr 25 2023 11:59PM. If you will need more time than this to complete your revisions, please reply to this message or contact the journal office at digitalhealth@plos.org. Please include the following items when submitting your revised manuscript:

We look forward to receiving your revised manuscript.

Kind regards,

Dukyong Yoon

Academic Editor

PLOS Digital Health

Journal Requirements:

1. Please send a completed 'Competing Interests' statement, including any COIs declared by your co-authors. If you have no competing interests to declare, please state "The authors have declared that no competing interests exist". Otherwise please declare all competing interests beginning with the statement "I have read the journal's policy and the authors of this manuscript have the following competing interests:"

2. Please ensure that Funding Information and Financial Disclosure Statement are matched.

3. In the Funding Information you indicated that no funding was received. Please revise the Funding Information field to reflect funding received.

4. Please provide separate figure files in .tif or .eps format only and remove any figures embedded in your manuscript file. Please also ensure that all files are under our size limit of 10MB.

5. We have noticed that you have uploaded Supporting Information files, but you have not included a list of legends. Please add a full list of legends for your Supporting Information files after the references list. 

Additional Editor Comments (if provided):

Reviewers' comments:

Reviewer's Responses to Questions

**Comments to the Author**

1. Does this manuscript meet PLOS Digital Health’s publication criteria? Is the manuscript technically sound, and do the data support the conclusions? The manuscript must describe methodologically and ethically rigorous research with conclusions that are appropriately drawn based on the data presented.

Reviewer #1: Yes

Reviewer #2: Yes

2. Has the statistical analysis been performed appropriately and rigorously?

Reviewer #1: Yes

Reviewer #2: Yes

3. Have the authors made all data underlying the findings in their manuscript fully available (please refer to the Data Availability Statement at the start of the manuscript PDF file)?

Reviewer #1: No

Reviewer #2: Yes

4. Is the manuscript presented in an intelligible fashion and written in standard English?

Reviewer #1: Yes

Reviewer #2: Yes

5. Review Comments to the Author

Reviewer #1: The authors trained an XGBoost machine learning model for patients with suspected COVID19 who presented themselves to Western Cape public hospitals during multiple waves of the pandemic. They found that the most important features for predicting adverse outcome were:

the requirement for supplemental oxygen, 

peripheral oxygen saturations, 

level of consciousness; and

age. 

These seem theoretically reasonably. Internal validation using a split-test sample showed convincing discrimination (C-statistic 0.91, 95% CI 0.90 to 0.91) and calibration (CITL of 1.05). The model achieved C-statistics of 0.84 (95% CI 0.84 to 0.85), 0.72 (95% CI 0.71 to 0.73) and 0.62, (95% CI 0.59 to 0.65) in the Western Cape Omicron wave, UK, and Sudanese external validation

cohorts. Calibration was sub-optimal with overprediction of adverse outcome across all risk

subgroups in Omicron cases (CITL of 1.7); and underprediction of adverse outcome in UK cases with

ancestral COVID19 infection and Sudanese patients with ancestral or Eta variant infection (CITL of

0.68 and 0.61 respectively). Results were not substantively changed in extensive sensitivity analyses

exploring different missing data mechanisms.

Although the article reads well, my sense is that it could use some very minor proofreading including a clarification of what is meant here by a 'retrospective' cohort (different authors tend to use that qualifier in ambiguous ways, unfortunately). It may also be worthwhile stating why some other machine learning baselines were not considered (including logistic regression) and whether there is a possibility of including them, or comparing them, to XGBoost.

Reviewer #2: The authors of this paper have conducted detailed ML and statistical analyses of COVID-19 patients and developed a model to predict adverse outcome early on during hospitalization. The analyses and interpretations are well-written but I have four main concerns with this manuscript:

1. Knowing that clinical course of patients with different COVID-19 variants is substantially different, it is hard to understand why the randomization for the model was done on the basis of temporality (and thus variants). It is conceptually difficult to comprehend how a model based on alpha/beta/delta variants can be generalized to an omicron infection. It is not clear why the train/test split was based on temporality. The model could have been (at least in theory) more accurate if it included representations from all variants and times. Even if the model was later retrained on the whole set, its ability to accurately account for the causal variant would be lower than if the original train/test split was completely random irrespective of time and variant.

2. In the same vein, a clinician's prediction (let alone that from an ML model), is unlikely to consistently hold across geographies, variants, times and host characteristics including host genetics. So I feel it is unfair to expect that the model will perform across external validations sets as selected for this study. Such a lack of generalization is not really overfitting but it is because of the disparity between train and test sets.

3. From the Supplemental Figure S1, it seems that the ED data got split roughly to 2:1 proportion. With a large dataset at hand, the authors should have chosen a larger training set. This would ensure that the model learns from a larger variety of clinical presentations. That would likely improve the model performance - both in internal and external validation.

4. My real difficulty is with the clinical value of this model. As the authors mention in discussion, the likelihood of such a model being accepted in clinical practice is low. Then, why the exercise? Given the black box nature of AI models that the authors allude to, this model is also less likely to provide an insight into how to predict adverse outcome. Thus, a model based on less representativeness of training, less accuracy in external settings and less acceptability in clinical practice seems to be only an academic exercise.

6. PLOS authors have the option to publish the peer review history of their article (what does this mean?). If published, this will include your full peer review and any attached files.

**Do you want your identity to be public for this peer review?** For information about this choice, including consent withdrawal, please see our Privacy Policy.

Reviewer #1: No

Reviewer #2: No

---

## [Decision Letter · Decision Letter 1]

27 Jun 2023

TRAINING AND TESTING OF A GRADIENT BOOSTED MACHINE LEARNING MODEL TO PREDICT ADVERSE OUTCOME IN PATIENTS PRESENTING TO EMERGENCY DEPARTMENTS WITH SUSPECTED COVID-19 INFECTION IN A MIDDLE-INCOME SETTING

PDIG-D-23-00004R1

Dear Dr Fuller,

We are pleased to inform you that your manuscript 'TRAINING AND TESTING OF A GRADIENT BOOSTED MACHINE LEARNING MODEL TO PREDICT ADVERSE OUTCOME IN PATIENTS PRESENTING TO EMERGENCY DEPARTMENTS WITH SUSPECTED COVID-19 INFECTION IN A MIDDLE-INCOME SETTING' has been provisionally accepted for publication in PLOS Digital Health.

Best regards,

Dukyong Yoon

Academic Editor

PLOS Digital Health

Reviewer Comments (if any, and for reference):

Reviewer's Responses to Questions

**Comments to the Author**

1. If the authors have adequately addressed your comments raised in a previous round of review and you feel that this manuscript is now acceptable for publication, you may indicate that here to bypass the “Comments to the Author” section, enter your conflict of interest statement in the “Confidential to Editor” section, and submit your "Accept" recommendation.

Reviewer #1: All comments have been addressed

2. Does this manuscript meet PLOS Digital Health’s publication criteria? Is the manuscript technically sound, and do the data support the conclusions? The manuscript must describe methodologically and ethically rigorous research with conclusions that are appropriately drawn based on the data presented.

Reviewer #1: Yes

3. Has the statistical analysis been performed appropriately and rigorously?

Reviewer #1: Yes

4. Have the authors made all data underlying the findings in their manuscript fully available (please refer to the Data Availability Statement at the start of the manuscript PDF file)?

Reviewer #1: No

5. Is the manuscript presented in an intelligible fashion and written in standard English?

Reviewer #1: Yes

6. Review Comments to the Author

Reviewer #1: The authors have addressed all of my concerns.

7. PLOS authors have the option to publish the peer review history of their article (what does this mean?). If published, this will include your full peer review and any attached files.

**Do you want your identity to be public for this peer review?** For information about this choice, including consent withdrawal, please see our Privacy Policy.

Reviewer #1: No
